# Vaginal and urinary evaluation of lactobacilli quantification by qPCR: Identifying factors that influence urinary detection and the quantity of *Lactobacillus*

**Youngwu Kim[1,2]*, Agnes Bergerat-Thompson[3], Caroline Mitchell[1,2]**

**1** Department of Obstetrics, Harvard Medical School, Gynecology and Reproductive Biology, Boston, MA, United States of America, **2** Division of Female Pelvic Medicine and Reconstructive Surgery, Vincent Obstetrics and Gynecology, Massachusetts General Hospital, Boston, MA, United States of America, **3** Massachusetts General Hospital, Boston, MA, United States of America

* ykim65@mgh.harvard.edu

**Data Availability Statement:** All relevant data are within the paper and its Supporting Information files.

## Abstract

Vaginal colonization with lactobacilli has been linked to the health of the lower urinary tract in women. There is growing evidence that the bladder has its microbiome related closely to the vagina. In this study, we compared the three common vaginal *Lactobacillus* species (*L. jensenii*, *L. iners* and *L. crispatus*) in vaginal and urine samples to identify factors that influence urinary detection and the quantity of *Lactobacillus*. We used quantitative real-time PCR (qPCR) assays to measure the concentration of *Lactobacillus jensenii*, *L. iners* and *L. crispatus* in paired vaginal swabs and clean-catch urine samples from pre-and post-menopausal women. We compared demographic variables and vaginal *Lactobacillus* quantity between women with vaginal detection of at least one of the three species, detection in both vagina and urine, or urine only. We performed Spearman correlation between vaginal and urinary quantities of each species. We used multivariable logistic regression models to determine predictors of detectable *Lactobacillus* species in both samples (vs. vagina only or urine only). Models were adjusted for variables selected a priori: age, BMI, condom use, and recent sexual activity. Ninety-three paired vaginal fluid, and urine samples were included in the final analysis. 44 (47%) had no detectable *Lactobacillus* species in their urine samples, and 49 (53%) had at least one of the three *Lactobacillus* species (*L. jensenii*, *L. iners* and *L. crispatus*) detected in urine. Most women were white (91.4%), with a mean age of 39.8 ±13.8 years. The two groups were similar in demographics, gynecologic history, sexual history, recent use of antibiotics or probiotics within 7 days of sample collection, Nugent scores, and urine-specific gravity. Among the three *Lactobacillus* species, *L. jensenii* was more commonly detected in urine than the other two. For all three species, detection in the urine sample alone was infrequent. The concentrations of all three species were higher in vaginal samples than in urine samples. For all three *Lactobacillus* spp., vaginal abundance was associated with the urinary abundance of the same species even after adjusting for the Nugent score. In Spearman correlation analysis, urinary and vaginal *Lactobacillus* concentrations were positively correlated within the same species, with the most significant

**Funding:** The authors received no specific funding for this work.

**Competing interests:** No authors have competing interests

correlation coefficient for *L. jensenii* (R = 0.43, p<0.0001). Vaginal quantities were positively correlated between the three species, as were urinary quantities to a lesser extent. There was no meaningful correlation between the urinary quantity of one *Lactobacillus* sp. and the vaginal quantity of another species. In summary, the vaginal quantity of *Lactobacillus* was the most significant predictor of concurrent detection of the same species in the bladder, confirming the close relationship between these environments. Strategies to promote vaginal *Lactobacillus* colonization may also bring urinary colonization and the health of the lower urinary tract.

## Introduction

Analysis of urine using molecular methods and enhanced culture techniques has identified a commensal community of bacteria in the bladder, an area once presumed to be sterile [1–3]. Bladder microbiota have been associated with a range of lower urinary tract disorders in women, including overactive bladder [4, 5], urgency and stress urinary incontinence [3, 6–8], painful bladder syndrome [9], and post-operative urinary tract infections [10]. Identifying determinants of bladder microbiota may offer opportunities to improve treatment for these conditions.

Urinary microbial communities differ in composition between men and women, suggesting that differences in anatomy impact the seeding of the bladder with bacteria [2]. The vaginal microbiome seems a likely source of female urinary microbiota. Among pre-menopausal women, the relative abundance of *Lactobacillus*, *Gardnerella*, *Prevotella*, *Ureaplasma*, and *Escherichia* species were correlated between vaginal and mid-stream voided urine samples [11]. In postmenopausal women, an expanded culture of catheter-collected urine samples demonstrated increased urinary *Lactobacillus* abundance after vaginal estrogen treatment [12].

Comparisons of the relative abundance of microbes in vaginal and urine samples using 16S rRNA sequencing have suggested a significant concordance between communities in the vagina and urine [11, 13]. This correlation was largely due to the high relative abundance of *Lactobacillus* in both vagina and the urine samples [11, 13]. However, this type of sequencing only measures relative quantities and may not detect lower abundance members of the community. Absolute measurement of individual species through qPCR offers a different assessment of the impact of a given species: a 20% relative abundance in a community of a thousand organisms may have a very different biological impact than 2% of a million organisms. In this study, we compared the absolute abundance of three common *Lactobacillus* species (*L. jensenii, L. iners and L. crispatus)* in vaginal and urine samples to identify the factors influencing urinary detection and the quantity of *Lactobacillus*.

## Materials and methods

### Study setting, population, and design

Women presenting for ambulatory gynecological care at Massachusetts General Hospital were recruited for an observational cohort study of vulvovaginal symptoms and provided informed consent (IRB # 2014P001066) between August 2014 and May 2017. Women were excluded if their age was under 21 years, if they were pregnant, had known HIV infection or immunosuppression due to illness or medication use, or were prisoners. All racial and ethnic groups were included.

Participants completed questionnaires about general health history, sexual history, recent antibiotics, vaginal product use, and the presence of vulvovaginal symptoms. All participants underwent pelvic examination and obtained a vaginal swab sample and a midstream clean catch urine sample. Paired vaginal and urine samples from 100 subjects were analyzed by quantitative PCR (qPCR) for the presence and abundance of three *Lactobacillus* species: *L. jensenii*, *L. iners* and *L. crispatus*.

## Sample collection and storage

Urine samples were stored at 4°C for < 24 hours and aliquoted into 1.8 mL cryovials for storage at −80˚C until thawed for analysis. Vaginal swabs were stored at −80°C until analysis. Gram-stained slides of vaginal fluid underwent evaluation using the Nugent criteria [14].

## Bacterial DNA extraction of vaginal and urine samples

Vaginal swabs were thawed at room temperature, re-suspended in 450μLof phosphate-buffered saline solution, then vigorously vortexed for 2 minutes, and the swab was removed [15]. To prevent crystallization in urine samples, after thawing at 4 C, Tris-EDTA (1M) was added at 10% of urine volume before collecting the sediments [16]. DNA from the vaginal swab and urine samples was extracted using DNeasy UltraClean Microbial Kit (QIAamp).

## Quantification of *Lactobacillus spp*. by quantitative real-time PCR (qPCR)

The QuantStudio™ 7 Flex System (Thermo Fisher Scientific) and Sequence Detection Software version 1.3 (Applied Biosystems) were used to perform and analyze all qPCR results. Species-specific primers sets, TaqMan probes, and thermocycler conditions are detailed in S1 Table. Each reaction was performed in a final volume of 20μL containing 10μL GoTaq® Master Mix 2X (Promega), 0.8μL of each primer, 3μL nuclease-free water, and 2μL template DNA as previously described [17]. Results are reported as 16S rRNA gene copies/swab, which were calculated using a plasmid standard curve ranging from 2.5 to $10^6$ copies/reaction. Each assay was run with water-blank control samples to assess for contamination.

The average values were used to calculate the final concentration per sample (i.e., per vaginal swab sample or per mL of urine). Samples detected below the lower limit of quantification (2.5 gene copies/reaction) or undetected were assigned the value of 0.5 copies/sample for logarithmic transformation. When none of the three *Lactobacillus* species were detected in either vagina or urine, the participant was excluded from the analysis.

## Statistics

Variables were compared between groups using an independent t-test (continuous), a Chi-square test, and Fisher's exact test (categorical). A linear regression was performed to assess the association between vaginal and urinary quantities of each species. Because the Nugent score is a rough approximation of the relative abundance of *Lactobacillus* vs. other bacterial morphotypes, we adjusted this analysis for the Nugent score to examine how an absolute quantity of vaginal lactobacilli impacts urinary quantity at a similar relative abundance. A multivariable logistic regression model was used to determine variables associated with the urinary detection of each *Lactobacillus* species. The model included the *a priori* selected variables which we hypothesized might impact urinary colonization with microbes: age, BMI, condom use, vaginal cleansing history, and vaginal *Lactobacillus* quantity. Age and menopausal status are highly correlated; thus, we only included age in the model. Spearman correlation was used to evaluate associations between vaginal and urinary quantities of all *Lactobacillus* spp. using

log$_{10}$-transformed 16S rRNA gene copy numbers. All analyses were conducted in Stata (16.1, StataCorp LLC, College Station, TX) and GraphPad Prism version 9 (GraphPad Software, Inc). A p-value <0.05 was considered statistically significant.

## Results

### Study population

Paired urine and vaginal samples were available for analysis from 100 women enrolled between August 2014 and May 2017. DNA was successfully extracted and analyzed from all specimens.

Ninety-three paired samples had at least one of the targeted *Lactobacillus* species (*L. jensenii*, *L. iners* and *L. crispatus*) detected in either vaginal or urine samples. Seven paired samples with no detectable quantities of any of the three *Lactobacillus* species in either sample were excluded from subsequent analyses. Of the remaining 93, 44 (47%) had no detectable *Lactobacillus* species in their urine samples, and 49 (53%) had at least one of the three *Lactobacillus* species (*L. jensenii*, *L. iners* and *L. crispatus*) detected in urine (Fig 1). The two groups were similar in demographics, gynecologic history, sexual history, recent use of antibiotics or probiotics within 7 days of sample collection, Nugent scores, and urine-specific gravity (Table 1). People with none of the three species of lactobacilli detected in urine had a higher BMI (p = 0.016)

Among the three *Lactobacillus* species, *L. jensenii* was more commonly detected in urine than the other two species (Fig 1). For all three species, detection in the urine sample alone was infrequent. The concentrations of all three species were higher in vaginal samples than in urine samples (Fig 2; S1 Fig) (p<0.001).

### Predictors of urinary detection of *Lactobacillus*

All three *Lactobacillus* spp., vaginal abundance was associated with the urinary abundance of the same species even after adjusting for the Nugent score (Table 2). When only the samples with quantifiable urinary lactobacilli were included in the analysis, the relationship remained true for *L. jensenii* only (S2 Table). Because there was a high proportion of urinary samples with undetectable or unquantifiable levels of *Lactobacillus* spp. in the urine, we created a binary categorical variable for the presence of each species in the urine sample. In

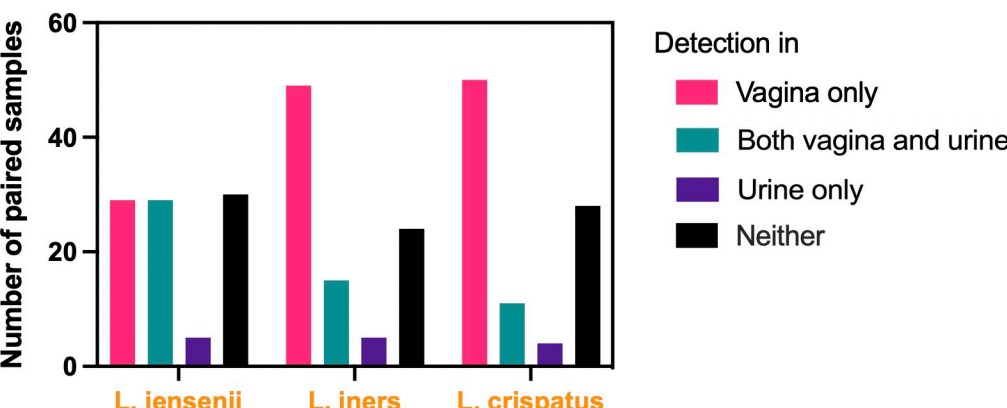

**Fig 1. Distribution and proportion of vaginal and urinary samples with detectable *L. jensenii*, *L. iners*, and *L. crispatus*.** Paired vaginal swabs and clean catch urine from 100 participants were assessed with species-specific qPCR for these three species of lactobacilli. Seven samples with no detection of any of the target species were excluded from the analysis.

**Table 1. Sociodemographic, health, and behavioral variables among women with detectable urinary *Lactobacillus spp*. versus none.**

| | | Detection of *L. crispatus*, *L. jensenii* or *L. iners* in urine | | |
|---|---|---|---|---|
| | | Not detected *N = 44* | At least one detected [a] *N = 49* | *p value* |
| **Age** | | | | |
| | Mean ± SD | 42.2 ± 14.1 | 37.4 ± 13.2 | 0.09 |
| **Race** | | | | |
| | White | 39 (89%) | 46 (94%) | 0.82 |
| | Asian | 3 (7%) | 2 (4%) | |
| | Black | 1 (2%) | 1 (2%) | |
| | Unknown | 1 (2%) | 0 (0%) | |
| **Body Mass Index (BMI)** | | | | |
| | Mean ± SD | 26.2 ± 6.4 | 23.4 ± 4.1 | 0.02 |
| **Postmenopausal** | | | | |
| | Yes | 14 (32%) | 10 (20%) | 0.21 |
| | No | 30 (68%) | 39 (80%) | |
| **Diabetes** | | | | |
| | Yes | 1 (2%) | 0 (0%) | 0.29 |
| | No | 43 (98%) | 49 (100%) | |
| **Education Background** | | | | |
| | High school | 2 (5%) | 1 (2%) | 0.67 |
| College or above | | 38 (86%) | 45 (92%) | |
| | Unknown | 4 (9%) | 3 (6%) | |
| **Reported annual income** | | | | |
| | less than $20,000 | 5 (11%) | 5 (10%) | 0.66 |
| | $20,000–50,000 | 9 (20%) | 11 (22%) | |
| | >$50,000 | 24 (55%) | 30 (61%) | |
| | Unknown | 6 (14%) | 3 (6%) | |
| **History of having more than two urinary tract infection in the past year** | | | | |
| | Yes | 7 (16%) | 6 (12%) | 0.61 |
| | No | 37 (84%) | 43 (88%) | |
| **Gender of current sexual partner** | | | | |
| | Male | 36 (82%) | 43 (88%) | 0.68 |
| | Female | 2 (5%) | 1 (2%) | |
| | Both | 1 (2%) | 2 (4%) | |
| | Unknown | 5 (11%) | 3 (6%) | |
| **History of sexual intercourse with at least one partner within 30 days prior** | | | | |
| | Yes | 12 (27%) | 25 (51%) | |
| | No | 28 (64%) | 21 (43%) | 0.07 |
| | Unknown | 4 (9%) | 3 (6%) | |
| History of hormonal contraception use[b] | | | | |
| | Yes | 9 (20%) | 13 (27%) | 0.72 |
| | No | 31 (70%) | 33 (67%) | |
| | Unknown | 4 (9%) | 3 (6%) | |
| **History of condom use** | | | | |
| | Yes | 4 (9%) | 10 (20%) | 0.13 |
| | No | 40 (91%) | 39 (80%) | |
| **Any antibiotic use within 7 days (vaginal or oral)** | | | | |
| | Yes | 7 (16%) | 5 (10%) | 0.41 |
| | No | 37 (84%) | 44 (90%) | |

(*Continued*)

**Table 1.** (Continued)

| | | Detection of *L. crispatus*, *L. jensenii* or *L. iners* in urine | | |
| --- | --- | --- | --- | --- |
| | | Not detected *N = 44* | At least one detected [a] *N = 49* | *p value* |
| **Any probiotic use within 7 days (vaginal or oral)** | | | | |
| | Yes | 9 (20%) | 14 (29%) | 0.37 |
| | No | 35 (80%) | 35 (71%) | |
| **Vaginal cleansing use within 7 days** | | | | |
| | Yes | 8 (18%) | 5 (10%) | 0.26 |
| | No | 36 (82%) | 44 (90%) | |
| Nugent score[c] | | | | |
| | Mean ± SD | 2.8 ± 2.2 | 2.5 ± 2.3 | 0.53 |
| **Urine specific gravity** | | | | |
| **Median (Interquartile range)** | | 1.02 (1.01–1.02) | 1.01 (1.01–1.02) | 0.07 |

[a] Any of the following *Lactobacillus* spp. detected by qPCR: *L.jensenii*, *L. iners*, *L. crispatus*

[b] defined yes, if reported use of Mirena, OCP, Nexplanon, Nuvaring[TM]; no, if surgical or natural menopause, surgical sterilization, rhythm method, Paraguard[TM], condom use or none.

[c]Nugent RP, Krohn MA, Hillier SL. Reliability of diagnosing bacterial vaginosis is improved by a standardized method of gram stain interpretation. J Clin Microbiol. 1991 Feb;29(2):297–301. doi: 10.1128/jcm.29.2.297–301.1991. PMID: 1706728; PMCID: PMC269757.

multivariable logistic regression analysis, the detection of *Lactobacillus* spp. in urine samples was significantly associated with the quantity of each *Lactobacillus* spp. in vaginal samples after adjusting for age, BMI, condom use, and vaginal cleansing history (Table 3). False detection rates were 13.56% (*L.jensenii)*, 13.70% (*L. iners)* and 16.67% (*L. crispatus)* when we used the probability threshold that maximizes Youden's J Index (Table 3). For *L. iners*, a history of recent condom use was also significantly associated with detecting that species in urine (Table 3). Additional chi-square analysis comparing binary presence/absence of each species in both urine and vaginal samples showed that while vaginal detection of *L. jensenii* was associated with urinary detection *(*p<0.001), this was not the case for *L. iners* and *L crispatus* *(*p>0.05*)* (S3 Table).

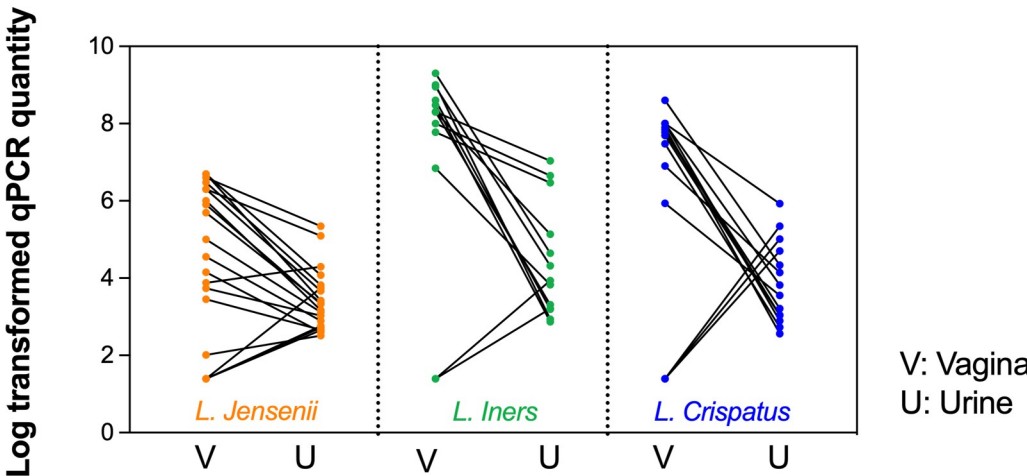

**Fig 2. Quantity of *L. jensenii*, *L iners* or *L. crispatus* detected in paired vaginal and urinary samples by species-specific qPCR.** Only pairs with the detection of the *Lactobacillus* species in the urine sample are depicted.

**Table 2. Association between vaginal and urinary quantities of each *Lactobacillus* species.**

|  | Regression coefficient[a] | 95% CI | P value |
|---|---|---|---|
| *L. jensenii* | 0.16 | 0.07–0.26 | 0.001 |
| *L. iners* | 0.16 | 0.07–0.25 | 0.001 |
| *L. crispatus* | 0.08 | 0.11–0.15 | 0.025 |

[a]For linear regression associating vaginal with urinary 16rRNA gene copies/sample for each species, adjusted for Nugent score (values $\log_{10}$ transformed for analysis)

## Correlation of *Lactobacillus* quantity between paired vaginal and urine samples

In Spearman correlation analysis, urinary and vaginal *Lactobacillus* concentrations were positively correlated within the same species, with the most significant correlation coefficient for *L. jensenii* (Fig 3. R = 0.43, p<0.0001). Vaginal quantities were positively correlated between the three species, as were urinary quantities to a lesser extent. There was no meaningful correlation between the urinary quantity of one *Lactobacillus* sp. and the vaginal quantity of another species.

## Discussion

In this study, we demonstrate a significant association between the vaginal quantity of three different *Lactobacillus* species and the detection of that species in a clean-catch urine sample. Using species-specific, quantitative PCR, we showed a correlation between vaginal and urinary quantities of the three species of lactobacilli: *L. jensenii, L. iners and L. crispatus*. This finding not only supports the growing evidence that there is an inter-relatedness between vaginal and urinary microbiomes but further reveals that the association is likely related explicitly to the absolute abundance of each *Lactobacillus* species.

Comparative studies of relative abundance using 16s rRNA sequencing and expanded quantitative urine and vaginal cultures have previously demonstrated significant overlap between vaginal and urinary microbiomes [3, 7, 11, 18]. Although the two methods are different, both identify a close association between the two microbiomes, primarily due to the predominance of the genus *Lactobacillus*[3, 7, 11, 18]. Specifically, the *Lactobacillus* species that

**Table 3. Multivariate logistic regression analysis for urinary detection of *L. jensenii, L. iners* and *L. crispatus*.**

|  | Urinary detection of *Lactobacillus* species | | | | | | | | |
|---|---|---|---|---|---|---|---|---|---|
|  | *L. jensenii* | | | *L. iners* | | | *L. crispatus* | | |
| Variables | OR | 95% CI | | *p* | OR | 95% CI | | *p* | OR | 95% CI | | *p* |
| Age | 1.01 | 0.98 | 1.05 | *0.48* | 1.00 | 0.96 | 1.05 | *0.83* | 0.97 | 0.91 | 1.03 | *0.36* |
| Body mass index | 0.94 | 0.86 | 1.04 | *0.24* | 0.92 | 0.81 | 1.05 | *0.23* | 0.88 | 0.75 | 1.04 | *0.14* |
| Condom use | 1.92 | 0.52 | 7.04 | *0.33* | 3.92 | 1.05 | 14.67 | *0.04* | 3.70 | 0.86 | 15.93 | *0.08* |
| Vaginal cleansing[a] | 0.42 | 0.07 | 2.37 | *0.32* | 1.41 | 0.29 | 6.93 | *0.67* | 0.52 | 0.04 | 6.07 | *0.60* |
| *Vaginal Lactobacillus*[b] quantity by qPCR | 1.54 | 1.18 | 2.00 | *0.00* | 1.28 | 1.04 | 1.56 | *0.02* | 1.31 | 1.04 | 1.65 | *0.02* |

OR: Odds ratio

CI: Confidence interval

[a] History of vaginal cleansing within 7 days of the evaluation.

[b] Specifically corresponding to the same species in the urine samples.

False detection rates calculated: *L.jensenii* (13.56%), *L. iners* (13.70%), *L. crispatus* (16.67%)

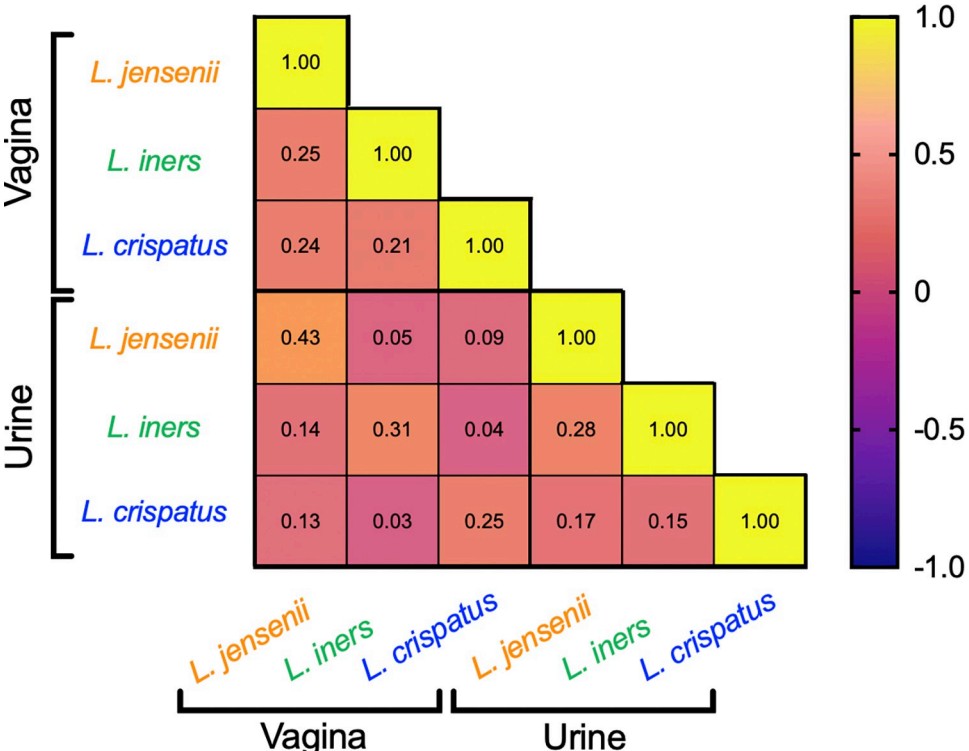

**Fig 3. Correlation between the urinary and vaginal quantity of *L. jensenii*, *L. iners* or *L. crispatus* measured by qPCR.** Each box is shaded according to the value of the Spearman correlation coefficient, and numbers represent the p-value.

showed the closest vaginal-urine correlation were *L. jensenii*, *L. iners* and *L. crispatus* [11]. Our use of quantitative PCR allowed us to demonstrate that the absolute amount of vaginal *Lactobacillus* spp., not just the relative amount, is an essential determinant of urinary colonization by the same species. Thus, even if there is a disruption to the microenvironment of vaginal flora, which causes a lowering of the relative abundance of lactobacilli, an individual may still have urinary colonization of the same species if the absolute abundance is sufficient.

In a recent study using 16s rRNA sequencing, the correlation between the relative abundances of vaginal and urinary *Lactobacillus* was similar between people with mixed urinary incontinence and a healthy control group [11]. The existence of this close relationship between urinary and vaginal *Lactobacillus* colonization, even in the control group, is additional evidence that many bacteria are likely to be shared between the vagina and bladder, not just by pathogenic organisms like *E. coli*. Whole genome sequencing of bacterial isolates from the vagina and bladder in 77 individuals revealed that strains of *L. iners* and *L. crispatus* isolated from an individual were the same in the two samples [18]. This is congruent with our finding that the effect of the vaginal abundance of *Lactobacillus* species on urinary abundance is species-specific. The abundance of one vaginal *Lactobacillus* species did not affect the abundance of other evaluated species in the urine. Knowing that *Lactobacillus* species have a diverse impact on lower urinary tract health, this species-specific association may play an essential role in creating the most effective treatment strategies.

Clinically, studies involving individuals with and without lower urinary tract symptoms have indicated that disruption of the genitourinary microbiome likely contributes to the

process of the pathogenesis of many symptoms [7, 19, 20]. A health-associated microbiome is characterized by decreased urinary bacterial diversity (price bladder, Adebayo), which usually means dominance by the genus *Lactobacillus*. Differences in the relative abundance of *Lactobacillus* species are associated with various lower urinary tract disorders (3,8,11,19). In a subgroup analysis of women younger than 51 years old, 16s rRNA sequencing of urine and vaginal samples showed that the control group had a higher *Lactobacillus* relative abundance [11]. The study recognized its limitations of having reported the associations mostly at the genus level. Our study, which demonstrated the association at the species level with the aforementioned qPCR method, demonstrated that increasing the vaginal quantity of *Lactobacillus* species increased the likelihood of detecting the same species in the urine.

In preserving vaginal health, the importance of symbiosis of $H_2O_2$-producing *Lactobacillus* species is well recognized [7, 21–23]. Maintaining a *Lactobacillus*-dominant vaginal microbiome is associated with lowered risks of sexually transmitted infections and vulvovaginal disorders, such as bacterial vaginosis and candidiasis, and improved obstetrical outcomes [21–25]. Similarly, microbiome studies of the bladder show an essential function of *Lactobacillus* species. For example, *L. crispatus*, universally accepted as health-promoting vaginal *Lactobacillus*, has also been established as an essential organism associated with urinary health, such as lowering the risks of urinary tract infections and urinary incontinence [3, 8, 12].

As mentioned above, our results strongly suggest species-specific associations between vaginal and urinary colonization, which implies that to promote urinary colonization with specific species, its vaginal quantity must be increased. This likely applies to other bacteria found in the genitourinary system as well. Exploring the species-specific effects of various bacteria in the bladder is essential. In a culture-based study, urinary *L. gasseri* was more commonly found in women with urinary urge incontinence [26]. In another study, *Atopobium vaginae were* associated with painful bladder syndrome [27]. Future species-specific studies involving a larger group of matched samples with and without lower urinary tract symptoms are warranted to provide further insight into developing effective preventative and curative therapies.

Although we did not find age or menopausal status to be an independent factor for urinary detection of three studied *Lactobacillus* spp., this is likely due to a lack of power, given the relatively narrow age range of participants, and the small number of postmenopausal participants. Aging has been shown to be a predictor of increased urinary microbiome diversity in several studies [20, 28]. Recently, a longitudinal study of 12 weeks of use of vaginal estrogen treatment in postmenopausal patients demonstrated a significant increase in urinary *Lactobacillus* spp. using the enhanced urine culture technique [12]. Although the study did not demonstrate increased vaginal quantities of *Lactobacillus* during the study period, it did show a significant decrease in the diversity within the microbial community, consistent with greater *Lactobacillus* dominance of vaginal communities [21, 24, 25, 29, 30]. Other studies with more extended follow-up periods have shown increased vaginal *Lactobacillus* with vaginal estrogen administration [29, 31], which likely contributes to more significant urinary *Lactobacillus* colonization and decreases UTI in women using topical estrogen preparations [32–34].

A major limitation of our study was the small number of relatively homogenous participants recruited for this study. Additionally, we did not have data on urinary symptoms from validated questionnaires, and we could not correlate and control for the presence of lower urinary tract disorders. Our study used mid-stream, clean catch samples rather than urine collected with a catheter; thus, some contribution of contamination during collection cannot be ruled out. However, the fact that we found urinary lactobacilli in people with no vaginal lactobacilli detected, and that associations were species-specific suggests that this was not a significant issue.

## Conclusion

Our results add to data suggesting that increasing the absolute quantity of vaginal lactobacilli will likely promote the prevalence of urinary *Lactobacillus* colonization. Additionally, our findings suggest vaginal colonization leads to direct seeding of the bladder since vaginal quantities were only correlated with quantities of the same species in the urine. Thus, any probiotic products for urinary health should contain species that will colonize the vagina.

## Supporting information

**S1 Fig. Quantity of *L. jensenii*, *L iners* or *L. crispatus* detected in paired vaginal and urinary samples by species-specific qPCR.** This figure includes samples without detectable *Lactobacillus* species in the urine sample.
(TIF)

**S1 Table. Primers, probe sequences, and PCR conditions for TaqMan assays.**
(DOCX)

**S2 Table. Association between vaginal and urinary 16S rRNA gene copy numbers of *L. jensenii*, *L iners* and *L. crispatus*, excluding samples without quantifiable detection in the urine.**
(DOCX)

**S3 Table. Association between vaginal and urinary detection of the same *Lactobacillus* species (N = 93).**
(DOCX)

## Acknowledgments

**Presentation:** This submission was presented as a short oral at the 2021 American Urogynecologic Society meeting on October 15, 2021, in Phoenix, Arizona.

## Author Contributions

**Conceptualization:** Youngwu Kim, Agnes Bergerat-Thompson, Caroline Mitchell.

**Data curation:** Youngwu Kim, Agnes Bergerat-Thompson, Caroline Mitchell.

**Formal analysis:** Youngwu Kim, Agnes Bergerat-Thompson, Caroline Mitchell.

**Funding acquisition:** Caroline Mitchell.

**Investigation:** Youngwu Kim, Agnes Bergerat-Thompson, Caroline Mitchell.

**Methodology:** Youngwu Kim, Agnes Bergerat-Thompson, Caroline Mitchell.

**Project administration:** Youngwu Kim, Agnes Bergerat-Thompson.

**Resources:** Agnes Bergerat-Thompson, Caroline Mitchell.

**Supervision:** Caroline Mitchell.

**Validation:** Caroline Mitchell.

**Writing – original draft:** Youngwu Kim.

**Writing – review & editing:** Youngwu Kim, Agnes Bergerat-Thompson, Caroline Mitchell.

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
