## [Decision Letter · Decision Letter 0]

29 Nov 2022

PONE-D-22-27069Vaginal and urinary evaluation of lactobacilli quantification by qPCR: identifying factors that influence urinary detection and the quantity of LactobacillusPLOS ONE

Dear Dr. Kim,

Thank you for submitting your manuscript to PLOS ONE. After careful consideration, we feel that it has merit but does not fully meet PLOS ONE’s publication criteria as it currently stands. Therefore, we invite you to submit a revised version of the manuscript that addresses the points raised during the review process.

 All three reviewers noted a number of issues that need to be more adequately addressed in a revised manuscript. These suggestions should greatly improve the clarity and presentation of the results.

We look forward to receiving your revised manuscript.

Kind regards,

Brenda A Wilson, Ph.D.

Academic Editor

PLOS ONE

Journal Requirements:

Additional Editor Comments:

Although all three reviewers found the study interesting, each noted a number of issues that if appropriately addressed would significantly improve the manuscript. I encourage you to revise your manuscript accordingly.

Reviewers' comments:

Reviewer's Responses to Questions

**Comments to the Author**

1. Is the manuscript technically sound, and do the data support the conclusions?

Reviewer #1: Yes

Reviewer #2: Yes

Reviewer #3: Yes

2. Has the statistical analysis been performed appropriately and rigorously? 

Reviewer #1: Yes

Reviewer #2: Yes

Reviewer #3: Yes

3. Have the authors made all data underlying the findings in their manuscript fully available?

Reviewer #1: Yes

Reviewer #2: Yes

Reviewer #3: Yes

4. Is the manuscript presented in an intelligible fashion and written in standard English?

Reviewer #1: Yes

Reviewer #2: Yes

Reviewer #3: Yes

5. Review Comments to the Author

Reviewer #1: Reviewer Report– Manuscript Number PONE-D-22-27069

Congratulations to the authors for a beautiful study and I believe that the present study should be published!

Said that, please find my general and minor comments below.

General comments

I recommended the present study for publication. However, the manuscript could be improved by the authors, and I made some suggestions in my report such as:

It is well-known that a midstream clean catch urine sample does not guarantee a sample without bacterial contamination from the penis or vagina from getting into a urine sample. This type of sample collection represents a main limitation of the present study; however, the authors recognized it at the end of the Discussion section of the study. Nonetheless, the number of women was also very low, and it should be mentioned in the limitations of the study.

The references and introduction could be improved (see my minor comments).

A better description of the Material and Methods section with a more exhaust explanation of the experimental design of this study must be realized by the authors.

The Discussion section of the manuscript needs to be improved by the authors, discussing better the detected lactobacilli species by themselves and as consortia in the vaginal microbiota and their potential role in the bladder microbiota/urinary microbiome (see my minor comments).

Minor comments

Abstract

Line 33- Please put “L. crispatus” in italic form.

Introduction

Lines 87-89- I invite the authors to clarify what type of significant concordance was found by DNA sequencing between vaginal and urine samples in the cited studies. This further information would be useful to the Readers. Likewise, more similar studies of urine samples could be added to obtain a better scenario for the Readers.

Materials and Methods

Lines 101-103- Plea rewrite the exclusion criteria sentence, more exactly: “Women were excluded if their age was under 21 years if they were pregnant, had known HIV infection or immunosuppression due to medication, or were prisoners.”

Lines 108-109- It is well-known that a midstream clean catch urine sample does not guarantee a sample without bacterial contamination from the penis or vagina from getting into a urine sample. This type of sample collection represents a main limitation of the present study; however, the authors recognized it at the end of the Discussion section of the study.

Line 118, 128, and 129- Please replace “ul” and “uL” with “µL”. Please check this type of error in the remaining manuscript.

Line 130- “Quantity was calculated using a plasmid standard curve”. Please briefly describe the standard curve for the Readers instead of just mentioning that it was previously described (what I think the authors are referring to the reference 17 in the previous sentence).

Lines 133-136- Please state the controls used in the qPCR assays. There is no mention or description of controls in qPCR assays, which is concerning for this reviewer.

Supplemental table 1- Please rectify the lack of units in some protocols of the “PCR conditions” column.

Results

Figure 1 does not add any useful information to the Readers. Better to add supplemental table 3 as a main table in the results.

In table 1- Please put “spp” in a non-italic form. Also, the “Not detected” column stated N= 44 but, in the race section, the authors only described 43 patients (39, 3, and 1).

Other mistakes were found such as different p-values, P values, and p values in main or supplemental tables or wrongly lactobacilli names (such as L. Jensenii and L. Crispatus in Supplemental table 3 and even in Figure 2 plus Supplemental Figure 1. So, I strongly recommend that the authors revised the tables/figures of the entire Results section.

Figure 2 and Supplemental Figure – The resolution is bad, and no quantitative values are clear to this reviewer. I suggest the authors amend the figures or figure out another way to illustrate these quantitative data.

Line 199- “…even after adjusting for the Nugent score (Table 2).” Please clarify how did you adjust the Nugent score.

Table 3- Please clarify how the authors evaluated the false discovery rate (FDR) in the multivariate logistic regression analysis for urinary detection of L. jensenii, L. iners, and L. crispatus. The evaluation of FDR in the multivariate logistic regression is not clear to me. Also, please put vaginal in the non-italic form at the “Vaginal Lactobacillus” line.

Discussion

Lines 243 and 270- Again, please put “spp.” In a non-italic form.

A better discussion is needed. Two pages are not enough to discuss the obtained results. The author generally discussed the Lactobacillus genus and briefly mentioned the role of L. gasseri. Also, the authors stated in lines 265-266 “L. crispatus has been well established as an essential organism in urinary health (3,8,11,12,20).” How L. crispatus is well established as an essential microorganism in the urinary tract? Explain to the Readers… because L. crispatus is well-known probiotic species in the vaginal tract preventing the colonization of opportunistic or primary pathogens and consequently urinary infections as well. The authors must discuss better his role in the urinary tract.

Also, probiotic activity is caused not only by individual Lactobacillus species but also by its multi-microbial interaction. Please read the recent publication (https://www.frontiersin.org/articles/10.3389/fcimb.2022.863208/full) and improved the discussion about L. iners, L. crispatus, and L. jensenii role in the vaginal microbiota and their potential role as part of the bladder microbiota/urinary microbiome.

In addition, the number of women was also very low, and it should be mentioned in the limitations of the study.

Congratulations on the present study. However, the manuscript needs obligatorily to be improved in order to be published. The Discussion and Material and Methods sections of the manuscript need to be improved by the authors.

Reviewer #2: This paper discusses the relationship between lactobacilli colonisation in the vagina and the health of the lower urinary tract in women. The results show that the microbiome of urine is closely related to the vagina and that the number of lactobacilli in the vagina is the most important predictor of the same species in the bladder, suggesting that strategies to promote lactobacilli colonisation in the vagina may be beneficial to the health of the lower urinary tract.

The results of this study provide support for the strategy of vaginal use of estrogen to improve urinary symptoms in postmenopausal women. Please describe what is the most recent contribution of this research work compared to the published literature on the subject.

Reviewer #3: A. Summary:

This study uses quantitative PCR to investigate vaginal and urinary colonization with prevalent vaginal Lactobacillus species.

B. Strengths:

1. This study helps fill an important gap in the literature. Recent and ongoing studies of the urinary microbiome have often demonstrated the finding of Lactobacillus species in the bladder. However, historically, the urinary microbiome field grew from studies using sophisticated, time-consuming and costly sequencing methodology on urinary samples only, not including paired vaginal samples, and many were conducted before investigations comparing urinary and vaginal colonization were undertaken. As in many areas of clinical-translational research, there are significant gaps in foundational research in the field of the urinary microbiome (urobiome) and this is one of those gaps. Addressing such gaps is not trivial, as the process often involves using older methods (not meeting NIH criteria for “innovation”) to establish principles that should have been investigated years ago. It might seem obvious that the source of at least some elements of the urinary microbiome in women is most often the vagina, but many studies of the urobiome have seemingly ignored this possibility, subsequently drawing illogical conclusions from the data, in some cases. The present study helps fill that gap in the literature with its simple and logical study design of comparing paired samples from these two anatomical sites in the same participants.

2. Other studies of the correlation between vaginal and urinary microbiomes are discussed in detail.

3. The basic study design is reasonable, with some exceptions (see below) and the molecular methods are appropriate.

4. Data presentation and discussion are clearly and insightfully presented.

C. Weaknesses:

1. The study appears to be part of a larger or parent study of vulvovaginal symptoms in women. It would be helpful to understand more about the symptoms experienced by the participants. The authors do note the inability to correlate the findings with lower urinary tract disorders as a weakness.

2. It would be helpful if the authors included more discussion of why they did not find that age/menopause was independently associated with urinary detection of the three studied Lactobacillus species.

6. PLOS authors have the option to publish the peer review history of their article (what does this mean?). If published, this will include your full peer review and any attached files.

Reviewer #1: **Yes: **António Machado

Reviewer #2: **Yes: **Ruifang Wu

Reviewer #3: No

---

## [Author Response · Author response to Decision Letter 0]

3 Feb 2023

Reviewer Report– Manuscript Number PONE-D-22-27069 -- ALSO included as a document in the attached files. 

Congratulations to the authors for a beautiful study, and I believe the present study should be published! 

Said that, please find my general and minor comments below. 

General comments 

I recommended the present study for publication. However, the manuscript could be improved by the authors, and I made some suggestions in my report such as: 

It is well-known that a midstream clean catch urine sample does not guarantee a sample without bacterial contamination from the penis or vagina from getting into a urine sample. This type of sample collection represents a main limitation of the present study; however, the authors recognized it at the end of the Discussion section of the study. Nonetheless, the number of women was also very low, and it should be mentioned in the limitations of the study

 [Thank you. This has been addressed in lines 322-326 “A major limitation of our study was…”]

The references and introduction could be improved (see my minor comments). 

A better description of the Material and Methods section with a more exhaust explanation of the experimental design of this study must be realized by the authors. 

The Discussion section of the manuscript needs to be improved by the authors, discussing better the detected lactobacilli species by themselves and as consortia in the vaginal microbiota and their potential role in the bladder microbiota/urinary microbiome (see my minor comments). 

Minor comments 

Abstract 

Line 33- Please put “L. crispatus” in italic form. [This has been changed]

Introduction 

Lines 87-89- I invite the authors to clarify what type of significant concordance was found by DNA sequencing between vaginal and urine samples in the cited studies. This further information would be useful to the Readers. Likewise, more similar studies of urine samples could be added to obtain a better scenario for the Readers. 

[Line 89 was added to add clarification as suggested. Now it reads “This correlation was largely due to Lactobacillus in both vagina and the urine samples”]

Materials and Methods 2 

Lines 101-103- Plea rewrite the exclusion criteria sentence, more exactly: “Women were excluded if their age was under 21 years if they were pregnant, had known HIV infection or immunosuppression due to medication, or were prisoners.” 

[Line 103-105 now reads, “Women were excluded if their age was under 21 years, if they were pregnant, had known HIV infection or immunosuppression due to illness or medication use, or were prisoners.] 

Lines 108-109- It is well-known that a midstream clean catch urine sample does not guarantee a sample without bacterial contamination from the penis or vagina from getting into a urine sample. This type of sample collection represents a main limitation of the present study; however, the authors recognized it at the end of the Discussion section of the study. [Unfortunately, yes, we recognized this as a major limitation also, and we elaborated on it further in the discussion section.]

Line 118, 128, and 129- Please replace “ul” and “uL” with “μL”. Please check this type of error in the remaining manuscript. [We addressed this. Thank you very much]

Line 130- “Quantity was calculated using a plasmid standard curve”. Please briefly describe the standard curve for the Readers instead of just mentioning that it was previously described (what I think the authors are referring to the reference 17 in the previous sentence). [This section has been in more detail as suggested. The revised lines 133-134 now read: “Results are reported as 16S rRNA gene copies/swab, which were calculated using a plasmid standard curve ranging from 2.5 to 106 copies/reaction.”]

Lines 133-136- Please state the controls used in the qPCR assays. There is no mention or description of controls in qPCR assays, which is concerning for this reviewer. [Thank you for this point. We have added the line 134-135: “Each assay was run with water-blank control samples to assess for contamination.”]

Supplemental table 1- Please rectify the lack of units in some protocols of the “PCR conditions” column. [This has been changed.]

Results 

Figure 1 does not add any useful information to the Readers. Better to add supplemental table 3 as a main table in the results. [We appreciate the Reviewer’s comment. Some people prefer data in a visual format, and some in a table format. Our preference is for a Figure, but we provided the Supplemental Table for those readers who prefer tables.] 

In table 1- Please put “spp” in a non-italic form. Also, the “Not detected” column stated N= 44 but, in the race section, the authors only described 43 patients (39, 3, and 1). [Reviewed this and updated the Tables]

Other mistakes were found such as different p-values, P values, and p values in main or supplemental tables or wrongly lactobacilli names (such as L. Jensenii and L. Crispatus in Supplemental table 3 and even in Figure 2 plus Supplemental Figure 1. So, I strongly recommend that the authors revised the tables/figures of the entire Results section. [Thank you for the comments. They have been changed.] 

Figure 2 and Supplemental Figure – The resolution is bad, and no quantitative values are clear to this reviewer. I suggest the authors amend the figures or figure out another way to illustrate these quantitative data. [Thank you very much. We have upgraded to resolve the resolution issues to meet the standards]

Line 199- “…even after adjusting for the Nugent score (Table 2).” Please clarify how did you adjust the Nugent score. [We adjusted the regression analysis using the Nugent scores that were obtained at the time of the sample collection.]

Table 3- Please clarify how the authors evaluated the false discovery rate (FDR) in the multivariate logistic regression analysis for urinary detection of L. jensenii, L. iners, and L. crispatus. 

[Because we considered the analysis for each Lactobacillus as its unique hypothesis, and thus we performed a single analysis per species and we did not adjust for multiple comparisons. We consider adjusting for multiple comparisons in gene expression studies or when looking for individual taxon associations in a broader microbiome study rather than in this more traditional regression analysis, including only a few variables.]

The evaluation of FDR in the multivariate logistic regression is not clear to me. [We have addressed this in Table 3]. Also, please put vaginal in the non-italic form at the “Vaginal Lactobacillus” line. 3 [We changed this]

Discussion

Lines 243 and 270- Again, please put “spp.” In a non-italic form.

A better discussion is needed. Two pages are not enough to discuss the obtained results. The author generally discussed the Lactobacillus genus and briefly mentioned the role of L. gasseri. Also, the authors stated in lines 265-266 “L. crispatus has been well established as an essential organism in urinary health (3,8,11,12,20).” How L. crispatus is well established as an essential microorganism in the urinary tract? Explain to the Readers… because L. crispatus is well-known probiotic species in the vaginal tract preventing the colonization of opportunistic or primary pathogens and consequently urinary infections as well. The authors must discuss better his role in the urinary tract. 

Also, probiotic activity is caused not only by individual Lactobacillus species but also by its multi-microbial interaction. Please read the recent publication (https://www.frontiersin.org/articles/10.3389/fcimb.2022.863208/full) and improved the discussion about L. iners, L. crispatus, and L. jensenii role in the vaginal microbiota and their potential role as part of the bladder microbiota/urinary microbiome.

[Thank you for the comments. We have significantly reworked our discussion section]

In addition, the number of women was also very low, and it should be mentioned in the limitations of the study.

Congratulations on the present study. However, the manuscript needs obligatorily to be improved in order to be published. The Discussion and Material and Methods sections of the manuscript need to be improved by the authors. [Thank you very much for your comments. We hope we were able to address your concerns]

Reviewer #2: This paper discusses the relationship between lactobacilli colonisation in the vagina and the health of the lower urinary tract in women. The results show that the microbiome of urine is closely related to the vagina and that the number of lactobacilli in the vagina is the most important predictor of the same species in the bladder, suggesting that strategies to promote lactobacilli colonisation in the vagina may be beneficial to the health of the lower urinary tract. 

The results of this study provide support for the strategy of vaginal use of estrogen to improve urinary symptoms in postmenopausal women. Please describe what is the most recent contribution of this research work compared to the published literature on the subject. [Thank you for your comments. I hope the new discussion portion answers your suggestions.]

Reviewer #3: A. Summary:

This study uses quantitative PCR to investigate vaginal and urinary colonization with prevalent vaginal Lactobacillus species.

B. Strengths:

1. This study helps fill an important gap in the literature. Recent and ongoing studies of the urinary microbiome have often demonstrated the finding of Lactobacillus species in the bladder. However, historically, the urinary microbiome field grew from studies using sophisticated, time-consuming and costly sequencing methodology on urinary samples only, not including paired vaginal samples, and many were conducted before investigations comparing urinary and vaginal colonization were undertaken. As in many areas of clinical-translational research, there are significant gaps in foundational research in the field of the urinary microbiome (urobiome) and this is one of those gaps. Addressing such gaps is not trivial, as the process often involves using older methods (not meeting NIH criteria for “innovation”) to establish principles that should have been investigated years ago. It might seem obvious that the source of at least some elements of the urinary microbiome in women is most often the vagina, but many studies of the urobiome have seemingly ignored this possibility, subsequently drawing illogical conclusions from the data, in some cases. The present study helps fill that gap in the literature with its simple and logical study design of comparing paired samples from these two anatomical sites in the same participants.

2. Other studies of the correlation between vaginal and urinary microbiomes are discussed in detail.

3. The basic study design is reasonable, with some exceptions (see below) and the molecular methods are appropriate.

4. Data presentation and discussion are clearly and insightfully presented.

C. Weaknesses:

1. The study appears to be part of a larger or parent study of vulvovaginal symptoms in women. It would be helpful to understand more about the symptoms experienced by the participants. The authors do note the inability to correlate the findings with lower urinary tract disorders as a weakness. [We agree with your thoughts. Future studies are needed to correlate symptoms with urobiome at the species level]

2. It would be helpful if the authors included more discussion of why they did not find that age/menopause was independently associated with urinary detection of the three studied Lactobacillus species. [This was also an unexpected finding for us as well. We have added on lines 310-312, “this is likely due to a lack of power, given the relatively narrow age range of participants, and the small number of postmenopausal participants.”]

---

## [Decision Letter · Decision Letter 1]

6 Mar 2023

Vaginal and urinary evaluation of lactobacilli quantification by qPCR: identifying factors that influence urinary detection and the quantity of Lactobacillus

PONE-D-22-27069R1

Dear Dr. Kim,

We’re pleased to inform you that your manuscript has been judged scientifically suitable for publication and will be formally accepted for publication once it meets all outstanding technical requirements.

Kind regards,

Brenda A Wilson, Ph.D.

Academic Editor

PLOS ONE

Additional Editor Comments (optional):

The reviewers' concerns appear to be adequately addressed.

Reviewers' comments:

Reviewer's Responses to Questions

**Comments to the Author**

1. If the authors have adequately addressed your comments raised in a previous round of review and you feel that this manuscript is now acceptable for publication, you may indicate that here to bypass the “Comments to the Author” section, enter your conflict of interest statement in the “Confidential to Editor” section, and submit your "Accept" recommendation.

Reviewer #1: All comments have been addressed

Reviewer #2: All comments have been addressed

2. Is the manuscript technically sound, and do the data support the conclusions?

Reviewer #1: Yes

Reviewer #2: Yes

3. Has the statistical analysis been performed appropriately and rigorously? 

Reviewer #1: I Don't Know

Reviewer #2: Yes

4. Have the authors made all data underlying the findings in their manuscript fully available?

Reviewer #1: Yes

Reviewer #2: Yes

5. Is the manuscript presented in an intelligible fashion and written in standard English?

Reviewer #1: Yes

Reviewer #2: Yes

6. Review Comments to the Author

Reviewer #1: The manuscript is very well-written, and the achieved results are very interesting to the Readers. Congratulations and thank you for the opportunity to read your recent research!

I already endorsed the publication. However, I just have one request:

Line 304 (or line 331 in the manuscript with track-changes)- Please replace the "Atopobium vaginae" with "Fannyhessea vaginae (previously known as Atopobium vaginae)", as you may consult in: https://www.ncbi.nlm.nih.gov/Taxonomy/Browser/wwwtax.cgi?mode=Info&id=82135

Reviewer #2: (No Response)

7. PLOS authors have the option to publish the peer review history of their article (what does this mean?). If published, this will include your full peer review and any attached files.

Reviewer #1: **Yes: **José António Baptista Machado Soares

Reviewer #2: No

---

## [Editor Report · Acceptance letter]

5 Apr 2023

PONE-D-22-27069R1 

Vaginal and urinary evaluation of lactobacilli quantification by qPCR: identifying factors that influence urinary detection and the quantity of *Lactobacillus*

Dear Dr. Kim:

I'm pleased to inform you that your manuscript has been deemed suitable for publication in PLOS ONE. Congratulations! Your manuscript is now with our production department. 

Kind regards, 

on behalf of

Dr. Brenda A Wilson 

Academic Editor

PLOS ONE